# Investigation of Structural Changes in Rectus Abdominis Muscle According to Curl-Up Angle Using Ultrasound with an Extended Field of View

**DOI:** 10.3390/ijerph192114525

**Published:** 2022-11-05

**Authors:** Chansol Park, Hwi-Young Cho, Chang-Ki Kang

**Affiliations:** 1Department of Health Science, Gachon University Graduate School, Incheon 21936, Korea; 2Department of Physical Therapy, College of Health Science, Gachon University, Incheon 21936, Korea; 3Neuroscience Research Institute, Gachon University, Incheon 21565, Korea; 4Department of Radiological Science, College of Health Science, Gachon University, Incheon 21936, Korea

**Keywords:** rectus abdominis, ultrasound, extended field of view, curl-up

## Abstract

The rectus abdominis (RA) muscle is related to abdominal muscle contraction and is divided into three sub-sections. In this study, changes in each section according to the curl-up angles were measured using ultrasound with an extended field of view (EFOV) scan and compared with the conventional scan. The results showed that the locations of the measurement position center were consistently detected only with the EFOV scan and that the change ratios in thickness and length between sections at the same angles were not significantly different, suggesting that each section was selectively activated. Furthermore, the ratios of the thickness and length changes in each section were significantly different between the rest position and curl-up angles. Specifically, in the section 2 there was a significant difference in the thickness and length changes between different angles, reflecting the characteristics of RA during contraction, and their variations were less than in the section 3. Therefore, the results suggest that changes in only a section of RA cannot be representative of overall RA activation, EFOV scan method can be useful for investigating the structural characteristics of RA in detail, and the proposed method can obtain various information about muscle contraction for efficient muscle treatment and muscle strengthening.

## 1. Introduction

The abdominal muscles, the main muscles of the trunk, are composed of the rectus abdominis (RA), external oblique (EO), internal oblique (IO), and transversus abdominis (TrA) muscles [1,2]. Abdominal muscles are associated with lumbar vertebral stability and pelvic motion [3,4,5] and are closely associated with body movements such as walking and posture maintenance [6,7]. Previous studies have also confirmed the link between the muscles used for breathing, speaking, and swallowing [8,9,10]. Accordingly, abdominal muscles play an important role in the body’s motor function and daily life movements [11,12]. However, damage to the abdominal muscles leads to an imbalance in the body. Long-term neglect leads to dysfunction-related diseases, such as decreased motor function and chronic back pain [13,14,15]. Diseases related to the abdominal muscles include sarcopenia and diastasis of the RA. Sarcopenia increases the incidence of diabetes due to insulin resistance [16,17] and diastasis weakens the abdominal muscles and reduces the function of the lumbopelvic region, resulting in pelvic dysfunction and low back pain [18,19]. Therefore, strengthening the abdominal muscles is essential to treat functional disorders, prevent injuries, and improve overall physical function [2,20]. Various modified exercise methods have been developed for this by changing postures and using additional devices [21].

Previous studies have demonstrated the effect of curl-ups by measuring muscle thickness and electrical signals when a curl-up is performed. Surface electromyogram (sEMG), a representative measurement method, was used to measure electrical signals generated by muscle contraction on the surface of the skin; however, the disadvantage is that it is difficult to measure the deep-seated muscles and visually confirm muscle movements [22,23]. In addition, noninvasive diagnostic images with high spatial resolution, such as computed tomography (CT) and magnetic resonance imaging (MRI), were used to visually identify the muscles. However, CT and MRI have the disadvantages of radiation exposure and high sensitivity to metals and movement, respectively. Furthermore, because of the closed structure of CT and MRI, it is difficult to apply the exercise method; limitations include a long scan time and high cost [24,25]. These limitations were overcome using ultrasound. Ultrasound is non-invasive, has no radiation exposure, is inexpensive, and can be used to check images in real time [24]. In addition, when compared to images acquired by CT and MRI, it had high reliability and validation and could be measured in a free posture. Therefore, it is suitable for measuring thickness, length, and volume changes according to structural changes of muscles while applying the exercise method and has been widely used in physical therapy and rehabilitation medicine [25,26,27]. The meta-analysis study related to the reliability and validity of the extended field of view scan (EFOV) provided the detailed understanding and its various applications [25].

Specifically, RA has been considered in various studies because it shows distinct changes when performing body movements. It is characterized by being divided into three sub-sections by linear alba and tendinous intersection [20,28]. The ultrasound image of the RA was obtained by placing the center of the probe on the side of the umbilicus level. The region of interest (ROI) is defined in the image’s center as the probe is moved laterally and then up and down at a constant rate [29,30]. However, because the brightness mode scan (B-mode) of sonography commonly acquires transducer-sized cross-sectional images, long and wide muscles need to be measured several times in parts [31], limiting the examination of muscle characteristics. Muscle contraction acts as a combination of several factors, such as length and thickness [32]. Additionally, the muscle’s shape and position changed depending on posture and angle [33]. Therefore, even if the probe is placed at the same point on the skin, the actual measured position changes owing to postural muscle contraction. Especially in muscles with large contractions, such as the RA, the variation in the measurement position would be much greater. To examine the movement of these muscles, it is necessary to check the area around the ROI using an EFOV with the panorama imaging technique of ultrasound. The EFOV can acquire a wide FOV image compared to the conventional B-mode and show superior precision and significance when compared with MRI images [34,35]. Information on the vertical length during muscle contraction is essential to confirm the structural characteristics of the RA along the vertical axis. But, it is difficult to obtain the information only with B-mode images, and few studies have been conducted to measure the vertical length of the RA using ultrasound. In addition, a previous study investigated the change according to the section of RA for two exercises such as trunk and leg lifting using EFOV, but not for curl-up angles [34]. Another study examined the change in EO, IO, TrA, and RA when a curl-up was performed [36]. However, the curl-up angles at 0°, 30°, 60° and 90° were examined in EO, IO, and TrA using EMG; RA change was observed in the thickness change using a B-mode.

Therefore, this study proposes an ultrasound imaging method to investigate the structural changes of the RA during contraction along the vertical axis, to compare the thickness and length changes in each section according to the angle during curl-up, and to compare the changes between sections at the same angle. Using the proposed method and knowledge, we can help improve body function and develop an efficient intervention and exercise method for preventing and treating RA-related diseases.

## 2. Materials and Methods

### 2.1. Participants

This study was approved by the Gachon University Research Ethics Board (1044396-202105-HR-105-01). Twenty-six healthy participants provided written informed consent before participation. They were 23.8 ± 2.21 (mean ± SD) years old males in their twenties who could easily perform a curl-up; they were 73.2 ± 8.68 kg in weight, 175.0 ± 6.01 cm in height, and had a 23.9 ± 2.35 kg/m^2^ body mass index.

The inclusion criteria were eligible participants to perform the curl-up protocol of this study. Exclusion criteria were subjects with a history of central or peripheral nerve lesions, abdominal and/or low back pain, medical rehabilitation, surgeries on the abdominal and/or spine, psychosomatic drugs, and psychiatric disorders [36,37].

### 2.2. Experiments

Before the experiments, the participants adjusted their position on the leg fixing strap for the supine hook-lying position of hip flexion 50° and knee flexion 90°, which is the rest posture of the curl-up; they were asked to place both hands behind the head [Figure 1].

After that, to confirm the starting scan point using EFOV, we used ultrasound (RS85, Samsung Medison, Seoul, Korea) and a linear transducer probe (LA2-14A, Samsung Medison, Seoul, Korea, frequency bandwidth: 2.0 to 14.0 MHz), providing the insertion and tendinous intersection of RA in B-mode images. The area between each tendinous intersection was divided into three sub-sections (S1, S2, and S3), downward from the insertion. Finally, for accurate comparison of the images obtained by B-mode and EFOV, an absorptive marker was attached 15 mm from superior, inferior, left and right of the center of the measurement point (CMP), corresponding to the superior 5 cm and lateral 5 cm from the umbilicus level [Figure 2].

The CMP was selected not to interfere with the participant’s curl-up performance and the researcher’s ultrasound scan during the pilot experiment. In addition, the distance between the absorptive markers was set within the scanning range of the ultrasound probe but did not interfere with the analysis of the acquired images. After curl-up was performed at each angle (20°, 40°, and 60°), ultrasound scanning was started. Each curl-up angle was adjusted using a medical electric bed and goniometer. When measuring, the upper plate of the bed was lowered about 5 degrees and participants practiced maintaining the posture in which their trunks did not touch the bed. Additionally, it was sufficient for the participants to hold the curl-up posture for 10 s after they were given sufficient explanation and practice about the curl-ups. The angles were performed randomly, except for the rest position (0°). Each curl-up angle was maintained for 10 s while holding breath and performing an ultrasound scan. The following measurements were conducted at each angle (except for the rest position): (1) B-mode conventional FOV scan for CMP (B-mode), (2) B-mode to confirm each tendinous intersection, (3) EFOV for each sub-sections (EFOV), and (4) caliper measurement of the vertical length of between each tendinous intersection. After a curl-up at each angle, there was a 30 s rest period between the measurements and a 2 min rest period between different angles.

An experienced radiologist conducted an ultrasound scan in which the transducer was held vertically to the skin for all scans. In the B-mode, the center of the probe was placed on the CMP of the RA; images were obtained containing left and right absorption markers attached to the skin surface at a 30 mm distance, which are shown at the top of the image [Figure 3A,C]. To measure the length of each section, the probe was moved horizontally downward starting from the insertion, the position of the tendinous intersection was marked, and the vertical length of the RA was measured using a caliper. In the EFOV, the probe was positioned perpendicular to the level of insertion of the RA at a point 50 mm lateral from the midline (along the longitudinal axis of the CMP) and scanned vertically from the insertion to S3; images were obtained with superior and inferior absorptive markers placed at 30 mm apart around the CMP [Figure 3B,D]. The same procedure was repeated for different curl-up angles.

### 2.3. Data and Statistical Analysis

The thickness and length of the RA were measured using the DICOM viewer software (RadiAnt Dicom Viewer, Medixant, Poznan, Poland, Central Europe). Two researchers independently performed measurements using the predefined guidelines, in which CMP was confirmed in the B-mode image and the thickness was measured by vertically connecting the superior and deep fascia of each section of the RA. In the EFOV image, the section on which the CMP was located was identified to examine the position change of the RA according to the curl-up angle. To examine the validity of the scan methods and the change in muscle thickness according to the angle, the thickness was measured by vertically connecting the superficial fascia to the deep fascia of the CMP at the thickest area of each section. In addition, to examine the validity of the EFOV and caliper, the vertical length of the section was measured by connecting each section’s upper and lower tendinous intersections. The ratio of changes in the thickness and length of each section was calculated using the following formula [27,36]:$$\mathrm{Ratio}=\frac{\mathrm{Thickness}\;\mathrm{or}\;\mathrm{Length}\;\mathrm{in}\;\mathrm{the}\;\mathrm{curl}\text{-}\mathrm{up}\;\mathrm{angle}\;}{\mathrm{Thickness}\;\mathrm{or}\;\mathrm{Length}\;\mathrm{in}\;\mathrm{the}\;\mathrm{rest}\;\mathrm{position}}$$

Statistical analysis was conducted using a statistical software (Jamovi version 2.2.5) to analyze the significance between the measurement methods (B-mode and EFOV) and the ratios of thickness and length changes according to the angle. The interclass correlation coefficient (ICC) was used for the reliability between two measurers, and a paired samples *t*-test was used for the difference in CMP thickness between B-mode and EFOV images and the difference in length of sections measured by EFOV images and calipers. The ratio of thickness and length changes of the RA sections obtained from the EFOV was compared with the ratio of change in each section at the same angle at the same section using one-way repeated measures analysis of variance (RM ANOVA). The mean difference was calculated by subtracting the means, e.g., $\bar{X_{1}}$ − $\bar{X_{2}}$, where the means of factors 1 and 2 are $\bar{X_{1}}$ and $\bar{X_{2}}$, respectively. The standard error (SE) was calculated by using the following formula: $\mathrm{SE}=\sqrt{SD_{1}{}^{2}/n_{1}+SD_{2}{}^{2}/n_{2}}$, where the standard deviations and sample sizes of factors 1 and 2 are *SD*_1_ and *SD*_2_ and *n*_1_ and *n*_2_, respectively. A sphericity test was performed to check for significant differences in the changes in RA. Greenhouse–Geisser-corrected results were used when the sphericity was not satisfied. The level of statistical significance was set at *p* < 0.05 throughout all statistical analyses after Tukey post-hoc test.

## 3. Results

Inter-observer reliability was assessed by comparing thickness and length measured by two researchers with B-mode and EFOV RA images, which was sufficiently high (Cronbach’s α = 0.997). The difference between the B-mode and the EFOV of CMP thickness was 11.60 ± 4.27 mm (t = 1.643, *p* = 0.113) at 0°, 14.75 ± 5.95 mm (t = 0.683, *p* = 0.501) at 20°, 15.40 ± 5.67 mm (t = −1.174, *p* = 0.251) at 40°, and 16.15 ± 6.60 mm (t = 0.965, *p* = 0.344) at 60°; no significant difference was noted between both the measuring methods. In addition, there were no significant differences between the sectional lengths measured using the EFOV images and calipers.

In the result of the position of CMP in the EFOV, the number of participants with CMP at 0°, 20°, 40°, and 60° was 22, 22, 21, and 20 in the S2, respectively. The others were located in the S3. Furthermore, at the rest position, 3, 13, and 10 participants in S1, S2, and S3 were found to have the thickest area, respectively. At 20°, 4, 16, and 6 participants were in S1, S2, and S3; 6, 14, and 6 at 40°; and 1, 18, and 7 at 60°, respectively. Both the researchers chose the same section for the thickest section.

The changing ratios according to the angle and the section were analyzed for the following: (a) the difference between sections at the same angle and (b) the difference between angles in the same section. The ratio of thickness change was not significantly different in each section. The ratio was 1.49 ± 0.08, 1.37 ± 0.04, and 1.33 ± 0.05 (F = 2.84, *p* = 0.068) at 20°; 1.52 ± 0.07, 1.50 ± 0.05, and 1.40 ± 0.05 (F = 1.85, *p* = 0.178) at 40°; and 1.46 ± 0.08, 1.51 ± 0.04, and 1.41 ± 0.06 (F = 1.16, *p* = 0.321) at 60° in the S1, S2, and S3, respectively. The ratio of length changes also showed no significant difference in each section. The ratio was 0.85 ± 0.04, 0.88 ± 0.02, and 0.84 ± 0.05 (F = 0.37, *p* = 0.693) at 20°; 0.82 ± 0.04, 0.80 ± 0.02, and 0.82 ± 0.04 (F = 0.103, *p* = 0.903) at 40°; and 0.80 ± 0.04, 0.78 ± 0.02, and 0.73 ± 0.03 (F = 1.77, *p* = 0.18) at 60° in the S1, S2, and S3, respectively.

The mean differences in the ratios of thickness change in each angle in the same section are summarized in Table 1 and Figure 4.

As shown in Table 1, there were significant differences between rest position and all angles in the S1, but not in between 20° and 40°, 20° and 60°, and 40° and 60°, where they were −0.03 ± 0.05 (*p* = 0.953), 0.03 ± 0.06 (*p* = 0.966), and 0.06 ± 0.05 (*p* = 0.712), respectively.

There were significant differences between the rest position and all angles in the S2. Furthermore, significant difference was presented in 20° and 40° (−0.13 ± 0.05, *p* = 0.044) and 20° and 60° (−0.15 ± 0.05, *p* = 0.018), but not in 40° and 60° (−0.02 ± 0.03, *p* = 0.937).

There were significant differences between rest position and all angles in the S3, but not at 20° and 40°, 20° and 60°, or 40° and 60°; they were −0.07 ± 0.05 (*p* = 0.456), −0.08 ± 0.06 (*p* = 0.576), and 0.00 ± 0.05 (*p* = 1.000), respectively.

The mean differences in the ratios of the length change in each section at the same angle are summarized in Table 2 and Figure 5.

As shown in Table 2, there were significant differences between the rest position and all angles in the S1, but not in between 20° and 40°, 20° and 60°, and 40° and 60°, where they were 0.03 ± 0.03 (*p* = 0.659), 0.05 ± 0.03 (*p* = 0.331), and 0.02 ± 0.03 (*p* = 0.836), respectively. 

There were significant differences between the rest position and all angles in the S2. Furthermore, significant differences were presented in 20° and 40° (0.08 ± 0.02, *p* = 0.005) and 20° and 60° (0.10 ± 0.02, *p* = 0.002), but not in 40° and 60° (0.02 ± 0.02, *p* = 0.702).

There were significant differences between the rest position and all angles in the S3. Furthermore, the differences in 20° and 60° and 40° and 60° were significant as 0.12 ± 0.03 (*p* = 0.011) and 0.09 ± 0.03 (*p* = 0.044), respectively, but not in 20° and 40°, where it was 0.03 ± 0.04 (*p* = 0.902).

## 4. Discussion

In this study, we investigated the structural changes and sectional activation according to the curl-up angle in the entire range of the RA using the EFOV of ultrasound.

Curl-up is used as a typical exercise method to strengthen abdominal muscles, and various studies have been conducted on the same. A previous study classified the curl-up angles as 30°, 60°, and 90° and reported that RA activity increased under 60° [25,36]. Therefore, in this study, we examined the angles below 60° (where RA activity begins to increase), such as 20°, 40°, and 60°, for more precise measurements of RA muscular change.

In this study, inter-measurer reliability for the measured values was substantially high (Cronbach’s α = 0.997). This is probably because the location of CMP was marked using absorbent markers, which improved the accuracy of the measurer. In addition, there was no measuring interference between the measurers, though a consensus was reached when choosing the thickest region. Furthermore, the measurement range was commercially limited to one decimal place, making further fine measurements difficult, which possibly resulted in a small difference between raters. By comparing the measurement methods, it was confirmed that there was no significant difference in CMP thickness between the B-mode and EFOV. The B-mode is used to acquire transverse images of ROIs to measure thickness, length, volume, etc., but not for longitudinal information. However, EFOV can provide a wide FOV image with additional longitudinal information, thereby allowing examination of the entire range of large muscles and their relationship with adjacent muscles. Meanwhile, there was no sectional difference in length between the EFOV images and calipers. Therefore, RA thickness and length measurements using an EFOV seem to be valid.

This study found the positional change of the CMP within the RA using EFOV images according to the curl-up angle. In the rest position, the CMP position was mainly confirmed in S2 but changed as the curl-up angle increased, especially in S3. This means that although the origin and insertion point of the RA are the same depending on the individual, there are differences in the size and length of the section; the measurement position of the muscle can change owing to changes in posture. These results imply that existing measurement methods have limitations in accurately reflecting changes in RA [29,30]. Similarly, a previous EMG-related study showed that the attached EMG patch sites differ from person to person due to muscle length, spine bending, and skin and fat movements [33,38], which also suggests that the measuring position of electrical signals in muscles can change with changes in posture.

In this study, we also found that the thickness and length changes in each section of the RA appeared differently depending on the angle (Table 1 and Table 2). First, the ratio of the change in thickness and length between sections at the same angle was not significantly different. However, the section with a high change ratio differed from the angles, suggesting that the activated region differed depending on the angle. This means that RA contraction manifests differently in different sections and is closely related to the characteristic innervation of the RA. The nerves of the RA are divided into sections and are supplied from the 6th and 7th thoracic nerves [39]. These innervations were consistent with the findings of a previous study which also showed selective activation of the section when the RA contracted [29,40].

In addition, we compared the ratio of thickness and length changes by angle within the same section. The results showed that the thickness of each section significantly increased at all angles compared with the rest position. This result was consistent with that of a previous meta-analysis [25]. However, there were no significant differences between 20° and 40°, 20° and 60°, and 40° and 60° in S1 and S3, and between 40° and 60° in S2. It is presumed that this is because the ratio of thickness change increases at a similar level in each section. In addition, we found a reduced length of each section at all angles compared to the rest position, which was also similar to the length variation in RA shown in previous studies [34]. In a previous study, the influence on contraction was examined but the effect of curl-up angles was not studied. The present results showed that there was no significant difference between the combinations of 20° and 40°, 20° and 60°, and 40° and 60° in S1; between 40° and 60° in S2; and between 20° and 40° in S3. This can also be explained by the ratio of the length change decreasing at a similar level in each section.

In this study, we demonstrated increased thickness and decreased length due to RA contraction as the curl-up angle increased; it had the same trend depending on the angle of each section. S1 and S3 also showed similar contractions at a curl-up angle of 20° or more, whereas S2 showed similar contractions at a curl-up angle of 40° or more. However, the ratio of thickness change substantially increased in S1 than it did in S3 as the angle increased. However, the ratio of length changes substantially increased in S3 than it did in S1 as the angle increased. These results might be due to the increased influence of the hip flexor muscles as the angle of curl-up increases [36]. Specifically, S3 is located closer to the hip flexor muscle so there is a limited change in the thickness when compared to S1. This is consistent with the previous finding that the upper abdomen on which S1 is located showed higher activity than the lower abdomen on which S3 is located as the angle increased [33]. In S1, the change due to contraction seems to be greater because it is thinner than the other sections. S3 could contract owing to the elevation of the upper body as the angle increased, while the lower body was relatively fixed. The presence or absence of an increase in thickness associated with muscle contraction could be affected by the activity of adjacent muscles. If adjacent muscles are less active, muscle contraction and thickness changes increase [41]. Therefore, in the S1, it can be explained that the degree of contraction in the S2 is much higher because it is attached to the ribs and is less active. In contrast, S3 is close to the pelvis and the adjacent muscles are hip flexor muscles whose activity increases as the angle increases. S2 reflected the properties of RA contraction in that it showed significant differences in the thickness and length changes between most angles, but S2 had lower thickness and length variations than S3. These results suggest that certain sections of the RA may not represent its overall activities.

B-mode is still an excellent way to view the cross-sectional information of specific body parts, but in the case of a wide muscle such as RA, repeated measures are necessary, increasing the fatigue of the participant. In addition, muscle contraction works in a complicated manner with various factors that can change the measurement region due to muscle movement by posture. Therefore, since repeated measurements are also accompanied by changes in posture, such as exercise, it should be avoided when measuring wide muscles. This limitation of B-mode can be solved using EFOV showing high concordance with a B-mode or MRI and evaluating more detailed and accurate structural changes in a wider area.

In the results of this study, it was confirmed that there is a difference in the activation of each section of the RA depending on the angle of curl-up. In addition, more effective exercise methods similar to curl-ups are also expected to be devised by confirming the structural change of RA according to the angle. This is because exercise methods and prescription are presented differently depending on the purpose and condition of people. Therefore, this study was the first study to investigate the sectional changes according to the curl-up methods. The muscular changes in execution methods, and/or curl-up angles and its variants, should be further investigated.

However, a limitation of this experiment is that it is essential to include the superficial fascia to the deep fascia in the FOV for RA thickness measurement using ultrasound. Therefore, participants with thick bellies were excluded from this study, so various body types were not considered. In addition, changes in the EO, IO, and TrA that make up the body trunk along with the RA were not investigated; muscle activity via EMG should be further measured. Furthermore, the difficulty in maintaining such an accurate angle remains a limitation, and additional methods or studies such as a device that can maintain posture are still needed to solve the problem. Lastly, this study has limitations in its application to the elderly or females because it targeted only young, healthy males. However, similar results are expected in healthy participants of different ages or gender. Furthermore, as a person ages, their motor capacity decreases, and individual muscle thickness may also be another variable. Therefore, there is a need to conduct further studies for various age groups and genders. In a future study, it should further analyze the changes according to the section of RA when performing various types of core exercises, including at different age and gender groups, and investigate the impact of these exercises for each section of RA. In addition, using EFOV to identify overall changes in muscles rather than simple changes in specific muscle points for each exercise, criteria for determining which parts of the muscle are most appropriate to measure when performing a specific exercise should be determined.

## 5. Conclusions

This study examined the characteristics of RA contraction along the vertical axis and analyzed the curl-up-related changes in the thickness and length of RA sections using ultrasound. In the result of this study, the greatest ratio difference of the thickness changes was at 0°–40° (−0.52 ± 0.07) of S1, at 0°–60° (−0.51 ± 0.04) of S2, and at 0°–60° (−0.41 ± 0.06) of S3 in each sub-section. The greatest ratio difference of the length change was at 0°–60° (−0.21 ± 0.04) of S1, at 0°–60° (−0.51 ± 0.04) of S2, and at 0°–60° (0.28 ± 0.03) of S3 in each sub-section. In this study, we used an EFOV that can more precisely measure structural changes in the RA by providing the changes in the sectional thickness and length that accompany changes in the curl-up angle. In summary, first, the results in the EFOV images showed a change in the measuring position due to the contraction of the RA. Second, the proposed method can accurately evaluate muscle contractions that act complicatedly with surrounding muscles. With these results, further development of exercise methods that reflect selective activation according to the angle of RA would be possible, and efficient exercise methods for muscle strengthening and for the prevention of injuries that may occur in RA could be further developed. Finally, the application of the measuring method can extend not only to the strengthening of abdominal muscles, including RA, but also to the development of interventions that are useful for the treatment and prevention of abdominal muscle diseases such as chronic low back pain, sarcopenia, diastasis rectus abdominis, and diabetes. This would also be applicable for developing mediation and establishing a treatment plan according to the degree of symptoms.

## Figures and Tables

**Figure 1 ijerph-19-14525-f001:**
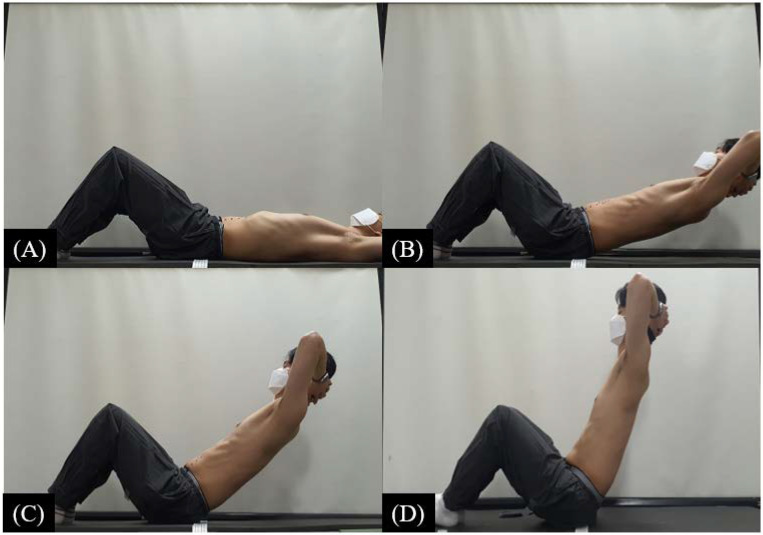
Curl-up posture by angle. (**A**) 0° (rest position) (**B**) 20° (**C**) 40° (**D**) 60°.

**Figure 2 ijerph-19-14525-f002:**
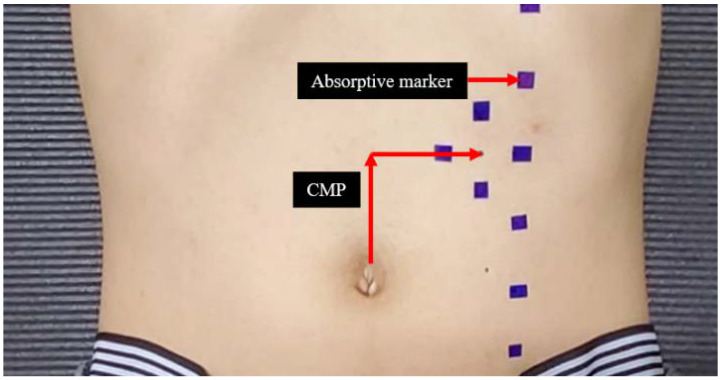
Center of measurement position (CMP) and absorptive markers.

**Figure 3 ijerph-19-14525-f003:**
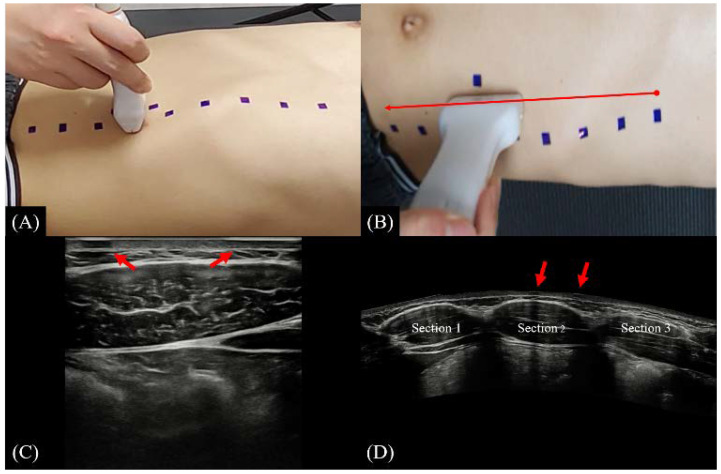
B-mode and EFOV scanning positions and representative images of each method. (**A**) B-mode scan; (**B**) EFOV scan; (**C**) Representative image of RA acquired using B-mode scan; (**D**) Representative image of RA acquired using EFOV scan. The arrow line in (**B**) is the direction of transducer movement during EFOV scanning. The red arrows in (**C**,**D**) indicate the shading images by absorptive markers. Abbreviations: RA, rectus abdominis; B-mode, brightness mode; EFOV, expanded field of view.

**Figure 4 ijerph-19-14525-f004:**
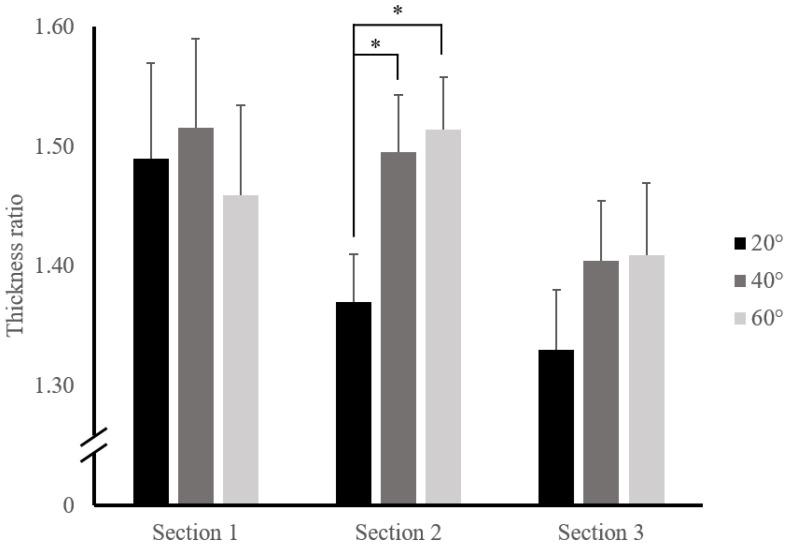
Group mean ratio of thickness changes by each curl-up angle in each section (the error bars indicate the standard error). Note that all curl-up angles showed significant differences in each section when compared to the rest position. * *p* < 0.05.

**Figure 5 ijerph-19-14525-f005:**
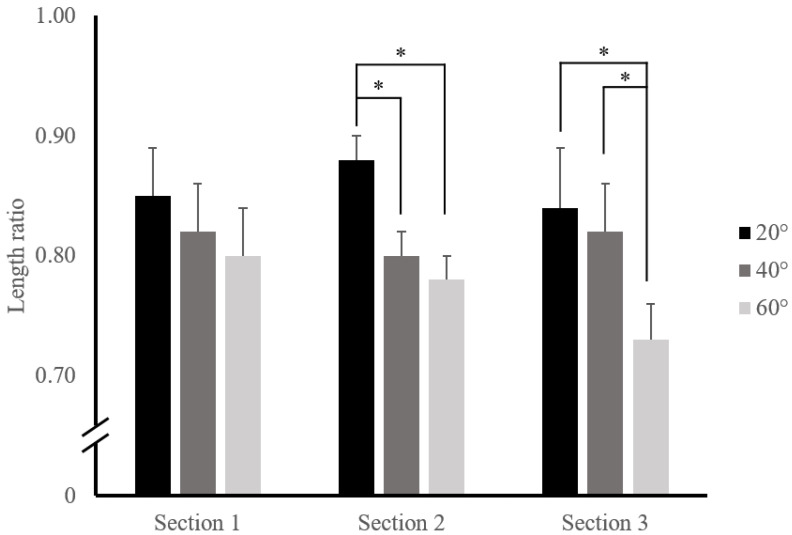
Group mean ratio of length changes by each curl-up angle in each section (the error bars indicate the standard error). Note that all curl-up angles showed significant differences in each section when compared to the rest position. * *p* < 0.05.

**Table 1 ijerph-19-14525-t001:** Comparison of the ratios of thickness changes by curl-up angle in each section.

	Angle 1	Angle 2	Mean Difference (Angle 1 − Angle 2)	Standard Error	*p*
Section 1	0°	20°	−0.49	0.08	<0.001 *
		40°	−0.52	0.07	<0.001 *
		60°	−0.46	0.08	<0.001 *
	20°	40°	−0.03	0.05	0.953
		60°	0.03	0.06	0.966
	40°	60°	0.06	0.05	0.712
Section 2	0°	20°	−0.37	0.04	<0.001 *
		40°	−0.50	0.05	<0.001 *
		60°	−0.51	0.04	<0.001 *
	20°	40°	−0.13	0.05	0.044 *
		60°	−0.15	0.05	0.018 *
	40°	60°	−0.02	0.03	0.937
Section 3	0°	20°	−0.33	0.05	<0.001 *
		40°	−0.40	0.05	<0.001 *
		60°	−0.41	0.06	<0.001 *
	20°	40°	−0.07	0.05	0.456
		60°	−0.08	0.06	0.576
	40°	60°	0.00	0.05	1

* *p* < 0.05.

**Table 2 ijerph-19-14525-t002:** Comparison of the ratios of length changes by curl-up angle in each section.

	Angle 1	Angle 2	Mean Difference (Angle 1 − Angle 2)	Standard Error	*p*
Section 1	0°	20°	0.15	0.04	0.003 *
		40°	0.18	0.04	<0.001 *
		60°	0.21	0.04	<0.001 *
	20°	40°	0.03	0.03	0.659
		60°	0.05	0.03	0.331
	40°	60°	0.02	0.03	0.836
Section 2	0°	20°	0.12	0.02	<0.001 *
		40°	0.20	0.02	<0.001 *
		60°	0.22	0.02	<0.001 *
	20°	40°	0.08	0.02	0.005 *
		60°	0.10	0.02	0.002 *
	40°	60°	0.02	0.02	0.702
Section 3	0°	20°	0.16	0.05	0.009 *
		40°	0.18	0.04	<0.001 *
		60°	0.28	0.03	<0.001 *
	20°	40°	0.03	0.04	0.902
		60°	0.12	0.03	0.011 *
	40°	60°	0.09	0.03	0.044 *

* *p* < 0.05.

## Data Availability

The data presented in this study are available upon request from the corresponding author.

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
