# Peer review of "Investigation of Structural Changes in Rectus Abdominis Muscle According to Curl-Up Angle Using Ultrasound with an Extended Field of View"

_ijerph, 2022, doi:10.3390/ijerph192114525_

Round 1
Reviewer 1 Report
The authors have investigated using EFOV imaging along with conventional ultrasound for characterizing structural changes in rectus abdominis (RA) muscle during curl-ups. Two observers participated considering the objectivity of interpretation. Overall, the method was stated well, and the results were presented in detail. There are adequate speculations and explanations for experiment findings. I think this manuscript can be accepted after minor revisions.
Some comments:
1. The experiment subjects are ~24-year-old male individuals. Do the authors expect there should similar findings for other age or gender groups?
2. Three angles, 20, 40, and 60 deg., were investigated and it stated the measurement precision is one decimal. However, it might be difficult for the study subject to maintain the same posture during the experiment for 10 seconds. What's the possible range of angular movements during measurement? This can possibly be a source of errors and should be discussed.
3. It would be helpful to include more discussions on how the measurement of structural changes in RA muscle can aid the further development of exercise methods.
4. Because there is no significant difference observed between EFOV and traditional B-scans, does this indicate implicitly or imply that traditional ultrasound is still efficient or valuable for RA muscle characterization?
Author Response
Authors’ Responses to Reviewers’ Comments:
Thank you very much for your valuable comments and suggestions. We carefully revised the manuscript following the reviewer’s recommendations accordingly. Point-by-point responses are as follows:
Comments and Suggestions for Authors:
-----------------------------------------------------------------------------------------------------
Reviewer #1:
The authors have investigated using EFOV imaging along with conventional ultrasound for characterizing structural changes in rectus abdominis (RA) muscle during curl-ups. Two observers participated considering the objectivity of interpretation. Overall, the method was stated well, and the results were presented in detail. There are adequate speculations and explanations for experiment findings. I think this manuscript can be accepted after minor revisions.
Some comments:
- The experiment subjects are ~24-year-old male individuals. Do the authors expect there should similar findings for other age or gender groups?
R1_1: Thank you very much for your valuable suggestions. We are sorry for any confusion to the reviewers about the age part described in the inclusion criteria, in which there was no specific age limit, but it was misrepresented in view of the age of the participants. We have corrected the inclusion / exclusion criteria in the Method section, and we have added discussion about other age or gender groups in the Discussion section as follows:
“Inclusion criteria were eligible participants to perform the curl-up protocol of this study. Exclusion criteria were subjects with a history of central or peripheral nerve lesions, abdominal and/or low back pain, medical rehabilitation, and surgeries on the abdominal and/or spine, psychosomatic drugs, and psychiatric disorders [36,37].”
“Lastly, this study has limitations in its application to the elderly or women because it was targeted only young healthy men. However, similar results are expected in the healthy participants of different age or gender. Furthermore, as a person ages, their motor capacity decreases, and individual muscle thickness may also be another variable. Therefore, there is a need to conduct further studies for various age groups and genders.”
- Three angles, 20, 40, and 60 deg., were investigated and it stated the measurement precision is one decimal. However, it might be difficult for the study subject to maintain the same posture during the experiment for 10 seconds. What's the possible range of angular movements during measurement? This can possibly be a source of errors and should be discussed.
R1_2: We used an electric bed to conduct the experiment, able to accurately adjust the curl-up angle to be performed. The participant maintained their posture, while the bed was lowered. In the figure, the plate was completely lowered to provide adequate information about the curl-up angle. In the preliminary tests, it was confirmed that although maintaining the position in the lower angle curl-up (e.g., 20o) was the most difficult, we used a method to adjust the angle of the participant's upper body using a medical electric bed and the sonographer did scan as quickly as possible. Nevertheless, re-measurements were performed after taking sufficient rest, when there was a lot of movement. We have added how to adjust curl-up angles more detail in the Method section and further limitation in the Discussion section regarding experimental procedures was added as follows:
“Each curl-up angle was adjusted using a medical electric bed and goniometer. When measuring, the upper plate of bed about 5 degrees was lowered and participants practiced maintaining a position where their trunks did not touch the bed. And it was sufficient for the participants to hold the curl-up posture for 10-sec after they were given sufficient explanation and practice about the curl-ups.”
“Furthermore, the difficulty in maintaining such an accurate angle remains a limitation, and additional methods or studies such as a device that can maintain posture are still needed to solve the problem.”
- It would be helpful to include more discussions on how the measurement of structural changes in RA muscle can aid the further development of exercise methods.
R1_3: We have discussed the relationship between the structural changes and development of exercise methods as the reviewer suggested as follows:
There are different types of modifications to the exercise methods, including curl-ups. This is because exercise methods and prescription are presented differently depending on the purpose and condition of people. Therefore, this study was the first study to investigate the sectional changes according to the curl-up methods, and it should be further investigated the muscular changes in execution methods, and/or curl-up angles and its variants.
- Because there is no significant difference observed between EFOV and traditional B-scans, does this indicate implicitly or imply that traditional ultrasound is still efficient or valuable for RA muscle characterization?
R1_4: We have added the content of the difference between EFOV and B-mode scan as follows:
B-mode is still an excellent way to view the cross-sectional information of specific body parts, but in the case of a wide muscle such as RA, repeated measures are necessary, increasing the fatigue of the participant. In addition, muscle contraction works in a complicated manner with various factors that can change the measurement region due to muscle movement by posture. Therefore, since repeated measurements are also accompanied by changes in posture, such as exercise, so it should be avoided when measuring wide muscles. This limitation of B-mode can be solved by EFOV showing high concordance with a B-mode or MRI can be used to evaluate more de-tailed and accurate structural changes in a wider area.

Reviewer 2 Report
The manuscript deals with a topic related to physical activity and health of young people, therefore the reviewer finds the topic relevant and useful. Muscle training preserves people's vitality into old age, prevents the accumulation of body mass around the abdomen, and so on. For this reason, the question posed may be interesting to a large number of readers.
The reviewer has the following questions, comments and recommendations:
1. Can the authors state the practical benefits of their study in more detail? The reviewer thinks that only one sentence on this topic is small. It is good to allocate a paragraph for this, for example, before the conclusion.
2. Why was the study only done on young people?
3. What results would be obtained if carried out on other age groups? What is the authors' assumption?
4. The authors have made a good introduction to the researched issues. However, the reviewer lacks an overview of research similar to that conducted by the authors. Are there any and can any comparison be made?
5. Inclusion criteria is age >20, but what is the upper limit? Is there one and what are the considerations?
6. Authors can include numerical data from the research done in their conclusion by referring to it.
7. The reviewer thinks it is good to allocate a short section on the future work of the authors in relation to the researched problem.
8. To indicate the source for the formula from line 167.
9. To specify the formulas by which the Mean Difference and the standard error are determined.
Author Response
Authors’ Responses to Reviewers’ Comments:
Thank you very much for your valuable comments and suggestions. We carefully revised the manuscript following the reviewer’s recommendations accordingly. Point-by-point responses are as follows:
Comments and Suggestions for Authors:
-----------------------------------------------------------------------------------------------------
Reviewer #2:
The manuscript deals with a topic related to physical activity and health of young people, therefore the reviewer finds the topic relevant and useful. Muscle training preserves people's vitality into old age, prevents the accumulation of body mass around the abdomen, and so on. For this reason, the question posed may be interesting to a large number of readers.
The reviewer has the following questions, comments and recommendations:
- Can the authors state the practical benefits of their study in more detail? The reviewer thinks that only one sentence on this topic is small. It is good to allocate a paragraph for this, for example, before the conclusion.
R2_1: We have added benefit of EFOV scan as follows:
“B-mode scan is excellent method to confirm the cross-section of a specific body part, but in the case of a wide muscle such as RA, it is cumbersome to repeat measurement, and repeated measurement may increase the fatigue of the researcher and the participant due to the increase in the number of measurements. In addition, muscle contraction works in a complicated manner with various factors that can change the measurement region due to muscle movement by posture. Therefore, repeated measurements should be avoided when measuring wide muscles that are accompanied by changes in posture, such as exercise. This limitation of B-mode scan can be solved by EFOV scan. EFOV showing high concordance with a B-mode scan or MRI can be used to evaluate more detailed and accurate structural changes in a wider area. In the results of this study, it was confirmed that there is a difference in the activation of each section of the RA depending on the angle of curl-up. In addition, exercise similar to curl-up are also expected to be able to devise more effective methods by confirming the structural change of RA according to the angle.”
- Why was the study only done on young people?
R2_2: There were no restrictions on age or gender when recruiting participants. However, since this study was conducted at a university, the participants were young who were very interested in exercises, but young women were relatively reluctant because it required the process of exposing the upper body.
- What results would be obtained if carried out on other age groups? What is the authors' assumption?
R2_3: We have discussed regarding other age or gender groups in Discussion section as follows:
Lastly, this study has limitations in its application to the elderly or women because it was targeted only young healthy men. However, similar results are expected in the healthy participants of different age or gender. Furthermore, as a person ages, their motor capacity decreases, and individual muscle thickness may also be another variable. Therefore, there is a need to conduct further studies for various age groups and genders.
- The authors have made a good introduction to the researched issues. However, the reviewer lacks an overview of research similar to that conducted by the authors. Are there any and can any comparison be made?
R2_4: We overviewed the following references in the manuscript for further understanding as follows:
“A meta-analysis study related to the reliability and validity of the EFOV scan provided the detailed understanding and various applications of EFOV scan [25].”
“And previous studies investigated the change according to the section of RA for two exercises such as trunk and leg lifting using EFOV and the change in EO, IO, TrA, and RA when curl-up was performed [34, 36]. However, the angles of curl-up of 0°, 30°, 60° and 90° were examined in EO, IO, and TrA using EMG, and RA change was observed in the thickness change by B-mode scan.”
- Inclusion criteria is age >20, but what is the upper limit? Is there one and what are the considerations?
R2_5: We are sorry for any confusion to the reviewers about the age part described in the inclusion criteria, in which there was no specific age limit, but it was misrepresented in view of the age of the participants. The age of participants ranged from 20 to 28 years old. We have corrected the inclusion / exclusion criteria in the Method section, and we have added discussion about other age or gender groups in the Discussion section as follows:
Inclusion criteria were eligible participants to perform the curl-up protocol of this study. Exclusion criteria were subjects with a history of central or peripheral nerve lesions, abdominal and/or low back pain, medical rehabilitation, and surgeries on the abdominal and/or spine, psychosomatic drugs, and psychiatric disorders [36,37].
- Authors can include numerical data from the research done in their conclusion by referring to it.
R2_6: We have added numerical data in Conclusion section as follows:
In the result of this study, the greatest ratio difference of the thickness changes was at 0°-40° (-0.52±0.07) of S1, at 0°-60° (-0.51±0.04) of S2, and at 0°-60° (-0.41±0.06) of S3 in each sub-sections. The greatest ratio difference of the length change was at 0°-60° (-0.21±0.04) of S1, at 0°-60° (-0.51±0.04) of S2, and at 0°-60° (0.28±0.03) of S3 in each sub-sections.
- The reviewer thinks it is good to allocate a short section on the future work of the authors in relation to the researched problem.
R2_7: We have added some future works in Discussion section as follows:
In a future study, it should further analyze the changes according to the section of RA when performing various types of core exercises, including at different age and gender groups, and investigate the impact of these exercises for each section of RA. In addition, using EFOV to identify overall changes in muscles rather than simple changes in specific muscle points for each exercise, criteria for determining which parts of the muscle are mostly appropriate to measure when performing a specific exercise should be determined.
- To indicate the source for the formula from line 167.
R2_8: We have added references [27,36] related to the formula in line 167 as the reviewer suggested.
- To specify the formulas by which the Mean Difference and the standard error are determined.
R2_9: We have added formulas for the Mean Difference and the standard error in Method section as follows:
The Mean Difference was calculated by subtracting the means, e.g., xÌ… 1 – xÌ…2 , where the means of Factors 1 and 2 are xÌ… 1 and xÌ…2, respectively, and the standard error (SE) was calculated by using the following formula: SE = √(SD12 /n12+SD22 /n22 ), where the standard deviations and sample sizes of Factors 1 and 2 are SD1 and SD2 and n1 and n2, respectively.
